# Mapping language analysis
# of comparative characteristics

Ben De Meester[0000−0003−0248−0987], Pieter Heyvaert[0000−0002−1583−5719],
Ruben Verborgh[0000−0002−8596−222X], and
Anastasia Dimou[0000−0003−2138−7972]

Ghent University – imec – IDLab,
Department of Electronics and Information Systems,
Technologiepark-Zwijnaarde 122, 9052 Ghent, Belgium
{firstname.lastname}@ugent.be

**Abstract.** RDF generation processes are becoming more interoperable,
reusable, and maintainable due to the increased usage of mapping lan-
guages: languages used to describe how to generate an RDF graph from
(semi-)structured data. This gives rise to new mapping languages, each
with different characteristics. However, it is not clear which mapping lan-
guage is fit for a given task. Thus, a comparative framework is needed.
In this paper, we investigate a set of mapping languages that inhibit
complementary characteristics, and present an initial set of comparative
characteristics based on requirements as put forward by the reference
works of those mapping languages. Initial investigation found 9 broad
characteristics, classified in 3 categories. To further formalize and com-
plete the set of characteristics, further investigation is needed, requiring
a joint effort of the community.

**Keywords:** Mapping Language · RDF graph generation

## 1 Introduction

RDF graph generation started as an ad-hoc process: hard-coded applications
generate specific RDF graphs, typically from specific data sources in specific
data formats. Shortly after, mapping languages were introduced, of which the
RDB to RDF Mapping Language (R2RML) is the first W3C recommended map-
ping language to generate RDF graphs from relational databases (RDBs) [1].
When a mapping language is used, the rules that specify the generation are de-
tached from the processor that executes them [8]. This improves interoperability,
reusability, and maintainability of the rules, the processor, and the entire RDF
graph generation process [13].

New mapping languages – or extensions to existing mapping languages –
are proposed to increase functionality or user-experience, and cater to different
use cases. For example, mapping languages were proposed extending W3C rec-
ommendations, such as R2RML and SPARQL. The RDF Mapping Language
(RML) is proposed as an extension of R2RML to support heterogeneous data

sources [7, 8], and SPARQL-Generate is proposed as an alternative mapping language, serialized using a modified SPARQL syntax [12].

This multitude of mapping languages allows to support more use cases. However, this multitude also defeats the mapping language's purpose: interoperability and reusability of current mapping rules and processors decrease. Currently, mapping rules are incompatible across processors of different mapping languages.

The differences between mapping languages are currently not presented in a unifying framework. Finding a suitable mapping language thus involves a laborious process where end-users need to investigate the characteristics of different mapping languages and interpret the differences. There is no clear interoperable description of (non-)functional characteristics of different mapping languages.

In this position paper, we investigate a set of mapping languages of which the reference works of those mapping languages claim complementary characteristics. We align and present an initial set of comparative characteristics as put forward by the reference works. These characteristics can then be used to compare different mapping languages against each other. This work is not meant to be complete, but to trigger discussion. Further and thorough investigation is required. After presenting our reference works in Section 2, we discuss the characteristics in Section 3 and conclude in Section 4.

## 2   Reference works

R2RML is the W3C recommended mapping language for describing RDF graph generation from a relational database (RDB) [1]. It presents a language specifically designed as mapping language, serialized in RDF. Other recommendations that allow to interpret a datasource as an RDF graph exist, such as JSON-LD [16] and CSVW [17]. These recommendations target a single datasource type. However, the need for supporting different datasource types for different use cases influenced the proposal of (i) other non-standardized mapping languages, (ii) extensions of existing mapping languages, and (iii) different notations.

We discuss RML(+FnO), an R2RML extension that supports heterogeneous datasources and data transformations [5, 7, 8]; xR2RML, an [R2]RML extension that supports collections and nested mappings [14, 15]; FunUL, an R2RML extension that supports data transformations [11]; SPARQL-Generate, an alternative mapping language based on SPARQL [12]; and YARRRML, an alternative notation to, a.o., RML [10]. Specifically, we discuss the references of Table 1, and the requirements they put forward.

This list is not exhaustive, however, we choose to discuss these works as their reference work(s) list a set of requirements, and these sets of requirements are (partially) complementary with those of the other mapping languages.

*RML* The RDF Mapping Language (RML) extends the R2RML recommendation to take into account heterogeneous data sources [7, 8]. Apart from allowing to specify how to generate the subject, predicate, object, and optionally graph resources, RML allows for specifying the logical source that describes the iteration over the data records (e.g., iterating over tabular data or JSON objects),

**Table 1.** The reference works discussed in this paper.

| Mapping language | Reference work(s) |
|---|---|
| (1) RML(+FnO) | [5, 7, 8] |
| (2) xR2RML | [14, 15] |
| (3) FunUL | [11] |
| (4) SPARQL-Generate | [12] |
| (5) YARRRML | [10] |

and specifying the data source that specifies the actual data connection (e.g., reading a local XML file or accessing a remote NoSQL endpoint). Later, RML was combined with the Function Ontology (FnO) [2, 4] to include arbitrary data transformations in the generation descriptions (RML(+FnO)) [5].

*xR2RML* An [R2]RML extension to include mapping functionalities targeted at hierarchical (NoSQL) data structures [14, 15]. These advanced functionalities include (i) nested term maps (for addressing hierarchical relationships), (ii) collections and lists, and (iii) combining multiple query languages (for addressing, e.g., a JSON record saved in an RDB).

*FunUL* An R2RML extension to include data transformations in the form of JavaScript snippets, and support for CSV data sources [11]. The presented work poses a set of eleven requirements for mapping languages, applied to generating RDF graphs from CSV files.

*SPARQL-Generate* An alternative mapping language using a SPARQL-like syntax to describe the mapping [12]. It has extensible support for different data sources and data transformations. The presented work poses seven functional and non-functional requirements.

*YARRRML* A notation using YAML[1] to provide a user-friendly way to describe mapping rules [10] which can be translated into RML (or other) mapping rules.

## 3 Characteristics

We discuss the following characteristics based on the aforementioned reference works, and their posed requirements. We further classify as non-functional, data source support, or functional characteristics (summarized in Table 2). For each characteristic, we state which reference work posed the original requirement.

### 3.1 Non-functional characteristics

Non-functional characteristics are in relation to the user of the mapping language. This user is either the end-user (creating the mapping), or the developer (integrating the mapping language processor into his/her application).

---

[1] https://yaml.org/

**Table 2.** Summarizing the characteristics of discussed mapping languages, numbered from 1–5. It is specified when the reference works of a mapping language claimed (or refuted) a certain characteristic of another mapping language. For example, the reference works of (2) – xR2RML – claiming that (1) – RML(+FnO) – has characteristic DS1 is noted as ✓(2). Statements claimed by the author of this paper are starred. YARRRML's characteristics except for NF1 and NF2 are taken from RML(+FnO), as YARRRML mappings can be tranlated into RML(+FnO) mappings.

| Language | NF1 | NF2 | NF3 | DS1 | | F1 | F2 | F3 | F4 |
|---|---|---|---|---|---|---|---|---|---|
| (1) RML(+FnO) | ✗(4) | ✓(4) | ✓(4) | ✓(2/3/4) | RDF CSV (3/4) XML (4) JSON (4) HTML (4) RDF (4) | ✓ | ✓(4) | ✗(2) | ✗* |
| (2) xR2RML | ✗(4) | ✓* | ✗* | ✓ | RDB NoSQL | ✓ | ✗* | ✓ | ✓ |
| (3) FunUL | ✗(4) | ✓* | ✓ | ~* | RDB CSV | ✓ | ✓ | ✗* | ✗* |
| (4) SPARQL-Generate | ✓ | ✓ | ✓* | ✓ | CSV XML JSON HTML Binary | ✓ | ✓ | ✓* | ✓* |
| $5  YARRRML | ✓ | ✓* | ✓(1*) | ✓(1*) | ✓(1*) | ✓(1*) | ✓(1*) | ✗(1*) | ✗(1*) |

*NF1: Easy to use by Semantic Web experts (from SPARQL-Generate/YARRRML)*
Whether the mapping language is "easy to use by Semantic Web experts" [12]
to describe the generation process or not. This characteristic is addressed by
SPARQL-Generate and YARRRML, which both provide a syntax that is either
"familiar" (SPARQ-like) [12] or "human-readable" (YAML) [10] to write rules.
The other mapping languages are described in RDF, without a specific notation.

*NF2: Based on Semantic Web standards (from SPARQL-Generate)* Whether the
mapping language "[integrates] with a typical semantic web engineering work-
flow" [12] or not, i.e., if it is related to existing standards. This characteristic is
fulfilled by all mapping languages, namely, they are related to R2RML, SPARQL,
or YAML.

*NF3: Fully covering the generation process (from FunUL)* Whether the map-
ping language "allows the serialization of the [generation] process for further
reuse" [11] or not: whether the generation description is fully covered by the
mapping language, or certain parts are hard-coded. xR2RML requires retrieval
of the data records as part of the hard-coded process. The other mapping lan-
guages allow to specify the connection to the physical data source.

## 3.2   Data source support characteristics

*DS1: Supporting heterogeneous sources (from RML/xR2RML/SPARQL-Generate)*
Whether the mapping language is focused on a single type of data source, or can
support multiple. FunUL targets tabular data sources. The other mapping lan-
guages have support for – and are extensible to – other data sources.

Although this is not tied to the mapping language itself, we further detail
which data sources are currently supported by the reference mapping language
processors.

## 3.3   Functional characteristics

*F1: Supporting general mapping functionalities (from FunUL)* Whether general
mapping functionalities are provided or not, namely, specifying "M:N relation-
ships", "literal to IRI", "vocabulary reuse", "data types", "named graphs", and
"blank nodes" [11]. These functionalities include (i) generating subject, predi-
cate, object, and (optionally) graph resources, (ii) joining data, and (iii) spec-
ifying the used ontology and datatypes. This is typically supported by most
mapping languages.

*F2: Extensible (from RML/FunUL/SPARQL-Generate)* Whether additional func-
tionalities can be added or not, to "allow data to be manipulated and trans-
formed" [11]. RML supports this and proves it by combining data descriptions
and FnO, FunUL allows including JavaScript snippets, and SPARQL-Generate
relies on SPARQL 1.1's extension functions.

*F3: Supporting nested hierarchies (from xR2RML)* Whether nested data records can be joined or not, to "map data elements from rows as well as structured values (nested collections [...])" [15]. This is different from being able to query a hierarchical dataset: you can query a hierarchical dataset to return (and join) only non-nested data records. Instead, this characteristic relates to whether the mapping language itself can handle hierarchical data structures or not. Except for xR2RML, all R2RML extensions use the tabular datamodel to join data. xR2RML also supports combining query languages when a nested data record has a different serialization than the parent data record. SPARQL-Generate is assumed to support this, as it extends SPARQL, a graph-based query language.

*F4: Supporting collections and lists (from xR2RML)* Whether the mapping language allows "to generate hierarchical values in the form of RDF collections or containers" [15] or not. xR2RML has built-in support, SPARQL-Generate allows to generate all triples required to generate compliant RDF lists.

### 3.4    Discussion

The difference in support of certain functional characteristics can help finding a mapping language that is suitable for a certain use case. However, other characteristics also influence the choice of a mapping language. For example, depending on the use case, a different mapping language notation might be preferred. On the one hand, a description in RDF can be less ambiguous, allowing more accurate analysis of the generation description. On the other hand, a mapping description serialized in RDF is arguably less user-friendly to edit and maintain by users.

Further investigation is needed into the difference between serialization and notation of mapping languages, and the underlying model and semantics they employ. For example, comparing SPARQL-Generate with YARRRML: the latter is designed to be human-friendly, and used to employ the semantics (and functionalities) as proposed by RML(+FnO). The former is based on the SPARQL language, but exhibits similar functionalities compared to RML(+FnO). RML, xR2RML and FunUL extend the model of R2RML, which is based on a tabular data model. SPARQL-Generate, following SPARQL, is based on a graph model.

Finally, it needs to be investigated on how to divide which characteristics apply to the mapping language and which apply to the processors of that mapping language. For example, a mapping language can be extended to a specific datasource, however, support of that datasource might not be easily achieved in the processor. Another example is the automatic generation of metadata of the generation process. This is easier when the mapping language is described in a machine-understandable format, i.e., RDF [3, 6].

## 4    Conclusion

In this position paper, we provide an initial investigation towards a comparative framework for mapping languages. A more systematic review is further required.

It is apparent that this effort cannot continue without support from the larger community. This work needs to be extended to consider a more complete range of existing mapping languages. Choices made in this work – specifically, the classification of characteristics into *non-functional, data source support, functional*, and the level of detail of different characteristics – need to be verified.

The resulting framework will allow for a clear division of which mapping languages support which characteristics, and allows for end-users to more easily find the right language for their use-case. More, development effort can be consolidated on missing features and filling the gaps, instead of spending time on re-developing existing functionalities.

Further formalization and comparison between mapping languages can be tested using a uniform set of test cases. Recent works are looking into test cases sets for specific mapping languages [9]. This work can be extended to provide a set of test cases across mapping languages. We expect this work to start a larger discussion, and to provide a basis for a more complete and accurate comparative framework. The recently started Knowledge Graph Construction Community Group can be a crucial driver for further investigation.

*Acknowledgements* The described research activities were funded by Ghent University, imec, Flanders Innovation & Entrepreneurship (VLAIO), and the European Union. Ruben Verborgh is a postdoctoral fellow of the Research Foundation – Flanders (FWO).

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
