# OpenReview forum: "Mapping languages: analysis of comparative characteristics"
_eswc-conferences.org/ESWC/2019/Workshop/KGB — KGB 2019_

### Official Review · ~Oscar_Corcho1 · 2019-03-23
**A good starting point for a characterisation of mapping languages**

**Rating:** 4
**Confidence:** 3

**Review:**

I think that this paper comes as a vision paper in a timely moment, when there are several mapping languages to generate RDF from heterogeneous data sources, some of which have stemmed out from existing W3C recommendations, such as R2RML and SPARQL. This heterogeneity is leading to practitioners/users to have some doubts on which option to use depending on their needs, and having a descriptive comparative summary may be useful for this purpose. At the same time, it will surely spark discussions during the workshop.

Therefore, as a workshop paper I think that it makes a lot of sense to accept it, so that it can be used as a starting point for discussion for the community, since surely there will be discussions on the current set of characteristics that have been presented as well as on potential new characteristics. For example, from my point of view there may be at least one further characteristic that I would like to include here, which is whether the mapping language allows for having a an engine that is able to do query translation or only materialisation (e.g., I assume that SPARQL-Generate, maybe, does not allow for that, and I am not sure about whether the others have any restriction on this). This is yet another aspect that I normally have to consider when selecting a mapping language.
In the same line, for me DS1 and DS2 may be basically combined into only one characteristic, I think, but I may be wrong. And I do not understand well why FunUL does not have a tick on DS1, even though all of them are tabular data sources.
Another potential interesting feature would be whether there is any formal semantics of the language (something important for any representation language).

As for the rest of the text, I am happy with the discussion in general, and only have some minor comments:
- On the first sentence, the authors comment that RDF graph generation started as an ad-hoc process. I would not agree completely with this claim, since early mapping languages like D2R or R2O appeared almost at the same time as the first proposals for RDF generation from RDBs.
- The choice of YARRML may be questioned, although I understand well the benefits that it provides. So I am happy with it to be included, but it may be questioned by others.

Typos:
- the which
- discuss following characteristics --> insert "the"?
- SPARQ-like --> SPARQL-like
- Reference 13 should have rdf and skos in uppercase

---

### Official Review · ~Aidan_Hogan1 · 2019-04-04
**Raises more questions than it answers**

**Rating:** 3
**Confidence:** 2

**Review:**

The paper discusses a scheme for categorising mapping languages from legacy sources to RDF. The authors motivate this scheme based on the observation that there is now a wide variety of mapping languages and tools available, and interoperability (or more generally, making sense of the options) is becoming an issue. They thus propose nine dimensions for comparing these languages and tools, grouped into three categories: non-functional, functional, and support for data source. They then categorise five tools according to these dimensions.

I am convinced that the paper will lead to interesting discussion at the workshop, and hence I lean towards accepting, but I do have some general concerns about the paper:

* The proposed scheme is really quite preliminary. While the authors readily admit this, at some point I feel they defer to the community or the future or both a little too much, such that their position becomes a bit blurred. As an example, while reading the paper I felt that the authors need to differentiate langauges from tools; while later in the paper the authors acknowledge this issue, they decide to postpone it, but I think this is really something core about the proposal that the authors should have clarified: are these categories for languages or are they for methodologies/interfaces/tools?

* Some of the dimensions proposed were a little unclear to me, while others may require more subjective judgement.
- NF1 seems very subjective; I understand that this is the nature of non-funcional requirements, but still, I think the first goal should be for a core categorisation that everyone can agree on, rather than one that becomes subjective and divisive.
- I found the title of NF2: "Integrates in a typical semantic web engineering workflow" to be vague; maybe just say "Is based on a Semantic Web standard?" if that's the criterion used?
- Again, the title of NF3: "Reusable" can be interpreted in many ways. Why not rather say: "Standard serialisation for exchange of mappings" if that's the criterion used?
- I found the discussion of F3 a little difficult to understand.

* I'm not 100% sure about motivation of such a categorisation: it seems a useful means, but to what ends? Will this lead to the survey of mapping languages/tools, or to a possible standardisation process, or something else? What is wrong with having multiple such tools doing their own thing? Why could this not be left to "survival of the fittest" in terms of adoption? These are some conceptual doubts I have about where this work is leading. (I guess the authors will remark that the categorisation could be useful in many such respects, but maybe the paper could be clearer on this point.)

In general, while the paper raises more questions than it answers, this can of course be seen as a good thing for a position paper at a workshop. Hence I lean towards accepting.

## Minor comments

* "the first time standardized by the W3C recommended RDB to RDF Mapping Languages" The sentence is not so clearly written, but if I understand the claim that RDB2RDF was the first mapping standard for W3C, this seems arguable given standards like XSLT, GRDDL, etc.

* "The rules of a mapping language cannot be processed by a processor of another mapping language." This claim is too strong. I understand the intent (that current langauges are not compatible across current tools) but it needs to be stated differently.

* Regarding Table 2, it is not clear what "Self-claimed statements" means; also the referencing format using (1), (4), etc., is not clear for me.

* The writing would benefit from some more attention to detail; while readable, there's frequent typos. As some examples:
 - "Mapping languages[:] analysis of comparative characteristics" (or maybe "Mapping language analysis of ...")
 - "RDF generation processes [are]"
 - "This leads to a rise" -> "This gives rise to"
 - "however, this -which- is not meant"
 - "no[n]-standardized"
 - "allows +for+ specifying"
 - "For [each] characteristic-s-, we state"
 - "Integrates [with] a typical ..." (twice)
 - "FunUL [targets] tabular data sources"
 - "and provide-s- a basis"
 - etc.: please revise in more detail.

---

### Decision · Program_Chairs · 2019-04-08
**Acceptance Decision**

**Decision:**

Accept

**Comment:**

This contribution is accepted for presentation at the KGB2019 workshop, and for inclusion in its proceedings.